# Antimicrobial Residue Accumulation Contributes to Higher Levels of *Rhodococcus equi* Carrying Resistance Genes in the Environment of Horse-Breeding Farms

**DOI:** 10.3390/vetsci11020092

**Published:** 2024-02-17

**Authors:** Courtney Higgins, Noah D. Cohen, Nathan Slovis, Melissa Boersma, Pankaj P. Gaonkar, Daniel R. Golden, Laura Huber

**Affiliations:** 1Pathobiology Department, College of Veterinary Medicine, Auburn University, Auburn, AL 36832, USAppg0001@auburn.edu (P.P.G.);; 2Large Animal Clinical Sciences, School of Veterinary Medicine and Biomedical Sciences, Texas A&M, College Station, TX 77843, USA; ncohen@tamu.edu; 3Hagyard Equine Medical Institute, Lexington, KY 40511, USA; 4College of Sciences and Mathematics, Auburn University, Auburn, AL 36849, USA; mdb0067@auburn.edu

**Keywords:** *Rhodococcus equi*, antimicrobial pollution, antimicrobial-resistant bacterial population dynamics, antimicrobial resistance selection and maintenance

## Abstract

**Simple Summary:**

The role of the environment as a reservoir and source of antimicrobial resistance (AMR) is not fully understood. In this report, we explore the consequences of historical antimicrobial use in horse-breeding farms on the accumulation of antimicrobial residues in the environment and its impact on *Rhodococcus equi* populations carrying antimicrobial resistance genes (AMRGs). Understanding these relationships is important because the environment is the main source of *R. equi* infections of young horses. Moreover, the higher concentration of transferable AMRGs in the soil contributes to the emergence of pathogenic organisms carrying multidrug resistance that can affect humans and animals. This report reinforces how farm practices can have One Health impacts and suggests future research to investigate ways to mitigate the spread of AMR.

**Abstract:**

Antimicrobial residues excreted in the environment following antimicrobial treatment enhance resistant microbial communities in the environment and have long-term effects on the selection and maintenance of antimicrobial resistance genes (AMRGs). In this study, we focused on understanding the impact of antimicrobial use on antimicrobial residue pollution and antimicrobial resistance (AMR) in the environment of horse-breeding farms. *Rhodococcus equi* is an ideal microbe to study these associations because it lives naturally in the soil, exchanges AMRGs with other bacteria in the environment, and can cause disease in animals and humans. The environment is the main source of *R. equi* infections in foals; therefore, higher levels of multidrug-resistant (MDR) *R. equi* in the environment contribute to clinical infections with MDR *R. equi*. We found that macrolide residues in the environment of horse-breeding farms and the use of thoracic ultrasonographic screening (TUS) for early detection of subclinically affected foals with *R. equi* infections were strongly associated with the presence of *R. equi* carrying AMRGs in the soil. Our findings indicate that the use of TUS contributed to historically higher antimicrobial use in foals, leading to the accumulation of antimicrobial residues in the environment and enhancing MDR *R. equi*.

## 1. Introduction

The World Health Organization has identified antimicrobial resistance (AMR) as one of the greatest threats to human health [1] and one of the top health challenges facing the 21st century [2,3]. Animal production is thought to be one of the main contributors to AMR because 80% of antimicrobial products sold worldwide are destined to be used in animals, and this use is estimated to continue to increase [4]. Antimicrobial residues and their metabolites can be shed into the environment via the feces and urine of animals treated with antimicrobials [5]; however, the risks of antimicrobial pollution and its consequences for the environment’s microbiome and the maintenance of AMR have not been fully elucidated. Studies have shown that even small concentrations of antimicrobials in the soil can drive the spread antimicrobial resistance genes (AMRGs) between bacteria [6]. Moreover, antimicrobial pollution affects microbes’ gene regulation [7,8,9] and quorum sensing [10], contributing to higher phenotypic variability and virulence [10].

*Rhodococcus equi* is a soil saprophyte commonly found in the environment of horse-breeding farms. This facultative intracellular pathogen is known to cause severe pneumonia in foals [11], but it can also affect humans [12] and other animal species [13]. Therefore, *R. equi* is an ideal organism to investigate the interconnections between antimicrobial use, the shedding of antimicrobial residues in the environment, the effects of antimicrobial pollution on the persistence of AMR, and the risks of infections in animals or humans in contact with the contaminated environment. In foals, *R. equi* infection occurs after inhalation of aerosolized virulent bacteria, which then invade and replicate within the alveolar macrophages, ultimately causing pneumonia [14]. *R. equi* infections are endemic at many horse-breeding farms. Consequently, horse-breeding farms rely on the use of thoracic ultrasonographic screening (TUS) to identify foals with subclinical pneumonia to prevent severe disease and death from this insidious disease [15]. This technique consists of performing TUS in young foals to early identify lung lesions presumptively caused by *R. equi* and initiate antimicrobial treatment before the development of clinical signs (during the subclinical phase). The detection of subclinical *R. equi* pneumonia using TUS has become a routine procedure at many horse-breeding farms in central Kentucky, USA. Because of the high prevalence of subclinical disease in horse-breeding farms, the use of macrolides and rifampicin, the most common treatment for subclinical (and clinical) pneumonia in foals [16], has increased.

In the past decade, the emergence of multidrug-resistant (MDR) *R. equi* isolated both from foals [17] and their environment [18] at horse-breeding farms has been reported, likely driven by the increase in antimicrobial use to prevent *R. equi* infections following the implementation of TUS. Since then, investigators have shown that foals subclinically affected by *R. equi* recover spontaneously without the need for antimicrobial treatment. It is expected that efforts to reduce antimicrobial use at horse-breeding farms may result in a decreased prevalence of MDR *R. equi,* reducing the risk of MDR infections in foals. However, MDR *R. equi* might persist in the environment through compensatory mutations [19] or by constant pressure due to the accumulation of antimicrobial residues [20] in the environment following the historical overuse of antimicrobials. If the persistence of *R. equi* carrying AMRGs and antimicrobials in the environment occurs at endemic farms, MDR *R. equi* infections in foals will likely continue to occur. The purposes of this study were (i) to compare rates of antimicrobial use, the prevalence of *R. equi* carrying AMRGs, and the concentration of antimicrobial residues at horse-breeding farms in Kentucky over time and (ii) to determine the effect of antimicrobial use and antimicrobial residue accumulation on the prevalence of *R. equi* carrying AMRGs found in the environment at horse-breeding farms.

## 2. Materials and Methods

### 2.1. Farm Selection

The horse-breeding farms located in Kentucky that previously participated in an epidemiological study conducted by our group [18] were re-enrolled for this study. From the original 100 farms enrolled in 2017, 83 agreed to participate in this study. Client consent for participation was obtained from each farm.

### 2.2. Questionnaire Collection

A survey was administered to each farm to gather information about farm demographics, such as location, number of foals born, acreage, number of years raising horses, and farm history pertinent to *R. equi* infections, such as antimicrobial treatment rates, incidence of pneumonia in foals, mortality, and management practices to prevent the disease. Questionnaires received as part of the study conducted in 2017 [18] contained information for years 2014 to 2017, and questionnaires collected in 2021 contained information for years 2018 to 2021.

### 2.3. Sample Collection

Samples were collected between June and July of years 2017 [18] and 2021. A total of 9 surface soil samples from 3 paddocks commonly used to house mares and foals in each farm were collected with a hand shovel that was cleaned and disinfected with alcohol wipes between farms. The soil samples from each location of the individual farms were evenly combined in a sterile collection bag for each respective farm, totaling a composite sample of approximately 140 g per farm. The collection bag was then sealed in a larger plastic bag to avoid cross-contamination between farms. The soil samples were stored at −80 °C before transportation to Auburn University and sample processing.

### 2.4. Sample Processing

#### 2.4.1. *R. equi* Quantification

Soil samples were thawed, and 1 g was measured in preparation for 10-fold serial dilutions using a 10 mM solution of phosphate buffered saline (Ultra-Pure PBS-Tween^®^ pH 7.5, VWR Life Science). The serial dilutions were then plated on a series of selective media for *R. equi* quantification. The series of agar plates consisted of NANAT agar [21], NANAT agar with erythromycin (8 μg/mL), and NANAT agar with erythromycin (8 μg/mL) and rifampicin (50 μg/mL) [18]. *R. equi* colonies from each plate were identified with characteristic colony morphology [22], and the colony-forming units (CFUs) were recorded. The colonies recorded from the NANAT agar plates represented the CFUs of the total population of *R. equi* in 1 g of soil. The colonies selected in the NANAT agar plates with added erythromycin with or without rifampicin represented the *R. equi* potentially carrying AMRGs in 1 g of soil. The *R. equi* colonies from the NANAT agar plates containing erythromycin and/or rifampicin were tested for the presence of the species-specific gene *choE* [22] and the presence of the virulence gene *vapA* using conventional PCR as previously developed [14,23]. The presence of macrolide AMRGs was determined by PCR amplification of *erm*(46) [24] and *erm*(51) [25] following previously developed protocols [24,25]. *R. equi* were classified as carrying AMRGs if either or both *erm*(46) and *erm*(51) were detected by PCR. ETEST^®^ (bioMérieux, Durham, NC, USA) strips were used to measure the minimum inhibitory concentration (MIC) for the following antimicrobials: azithromycin, clarithromycin, clindamycin, doxycycline, erythromycin, quinupristin–dalfopristin, rifampicin, tetracycline, trimethoprim–sulfamethoxazole, and vancomycin, according to manufacturer information. Inocula from a direct colony suspension in accordance with the CLSI guidelines [26] were used at a concentration of 1 to 5 × 10^5^ CFUs. The concentrations tested for each antimicrobial ranged from 0.016 μg/mL to 256 μg/mL, and the control strains used to concomitantly test susceptibility were *Staphylococcus aureus* ATCC 29213 and *Enterococcus faecalis* ATCC 29212 [27].

#### 2.4.2. Antimicrobial Residue Quantification

The presence of antimicrobial residue in the soil samples for each farm and year was evaluated through antimicrobial extraction and mass spectrometry. Antimicrobial extraction was modified from a previously described protocol [28]. Briefly, 1 g of farm soil was combined with 3 mL of methanol in a 15 mL conical tube and mixed well for 10 min. The sample was then wrapped in foil to avoid light degradation and placed in a shaking incubator overnight at room temperature. After incubation, the sample was centrifuged at 10,000× *g* for 30 min. The supernatant was placed into a glass vial and dried down with nitrogen gas dispensed from an evaporator (Reacti-Vap^®^, Thermo Fisher Scientific, Pittsburgh, PA, USA) in a dark room. The dried sample was reconstituted with 150 μL of a 50% methanol solution and vortexed for 5 min. Once reconstituted, the sample was transferred to a 2 mL microcentrifuge tube and centrifuged at 10,000× *g* for 30 min. The supernatant was then transferred to a new 2 mL microcentrifuge tube. The reconstituted samples were then submitted for mass spectrometry. Standard curves were constructed for each antimicrobial of interest by spiking a known gradient concentration into sterile soil. Standards were purchased commercially, and specifications for each antimicrobial are included in Appendix A.

The analysis was performed on a Vanquish UHPLC system (Thermo Fisher Scientific, Pittsburgh, PA, USA) coupled with a quadrupole orbitrap mass spectrometer (Orbitrap Exploris 120, Thermo Fisher Scientific, Pittsburgh, PA, USA) with electrospray ionization (H-ESI using Xcalibur software, V4.4.16.14). An injection of 10 μL of the sample or standard was made on a C18 column (SymmetryShield RP18, 3.5 µm, 4.6 × 150 mm, Waters Corporation, Milford, MA, USA) held at room temperature with a 1 mL/min flow rate of mobile phase solution A (99.9% water with 0.1% formic acid) and solution B (99.9% methanol with 0.1% formic acid). The gradient began at 25% B, held for 2 min, followed by a linear ramp to 65% B at 7 min and 95% B at 8 min, held for 1 min, then back to 25% B for a total analysis time of 14 min. The flow was diverted to waste for the 1.8 min of analysis and at 9 min. The mass spectrometer utilized positive mode for the entire analysis time, and negative mode was used from 8.3 to 9.3 min. The positive mode scan range was 100–1000 *m*/*z* with a resolution of 120,000, a standard AGC target, a 70% RF lens, and a maximum injection time of 100 ms, with mild trapping enabled and EASY-IC on at the run start. The negative mode was used for the analysis of 2,5-dichlorothiophene-3-sulfonamide, and the resolution was set to 60,000 and the maximum injection time was set to auto; all other parameters were the same as the positive mode settings. The spray voltage was 3400 V in positive mode and 2800 V in negative mode; the ion transfer tube temperature was 350 °C; and the vaporizer temperature was 320 °C. The processing method required a mass tolerance of less than 5.1 ppm.

### 2.5. Data Analysis

Data were analyzed using descriptive and inferential methods. For descriptive purposes, categorical data were presented as fractions and proportions, and non-normally distributed data were presented as medians and ranges. For inferential purposes, generalized mixed-effects modeling was used to evaluate the effect of macrolide residues and year on *R. equi* carrying AMRGs in soil samples with the farm identification number as the random factor to account for repeated measurements (2017 and 2021). Generalized mixed-effects modeling was also used to evaluate the effect of the use of TUS on the concentration and presence of macrolide residues in soil samples with the farm identification number as the random factor to account for repeated measurements (2017 and 2021). When an outcome variable was continuous (log-transformed *R. equi,* antimicrobial residue concentration), negative binomial mixed-effects modeling was used; when an outcome variable was categorical (presence or absence of *R. equi* carrying AMRGs, presence or absence of antimicrobial residues), logistic mixed-effects modeling was used. No interaction terms were included in the best fit model. Odds ratios (ORs) were calculated by exponentiating model coefficients and standard errors (SE). Paired data with non-parametric distributions were compared using the Wilcoxon signed-rank test. For all analysis, statistical significance was set to *p*-value < 0.05.

## 3. Results

### 3.1. Descriptive Data from Questionnaires

Of the 83 farms included in the study, 86% (71/83) completed a questionnaire in 2017, and 29 out of these 71 farms (41%) completed a questionnaire in 2021. A loss of follow-up was associated with the presence of *R. equi* carrying AMRGs; therefore, the questionnaire data are only represented descriptively. Because TUS is a variable of high interest based on previous epidemiological studies [29], we contacted farms that did not originally respond to a questionnaire in 2021 and were able to collect data about TUS practices for further inferential analyses. This second follow-up effort yielded complete paired data for TUS from 56 farms. When including only farms that fully responded to the questionnaires in both 2017 and 2021 (29 farms), the proportion of foals treated with antimicrobials and the number of farms treating foals with antimicrobials increased from 10 to 12.4% and from 66 to 79%, respectively (Table 1). The density of horses and foals per acre decreased slightly, and the mortality of foals from any cause remained virtually inexistent (Table 1). Considering the 56 farms that provided data regarding TUS from both years, the number of farms using TUS decreased from 77% (43/56) in 2017 to 66% (37/56) in 2021 (Table 1). Most farms (57%; 32/56) reported using TUS during the preceding decade, 14% (8/56) reported not using TUS in the preceding decade, 27% (15/56) reported using TUS in the period between 2014 and 2017 but not since 2017, and 2% (1/56) indicated they started using TUS after 2017 (Table 1).

### 3.2. Prevalence of R. equi Carrying Antimicrobial Resistance Genes (AMRGs)

Of the 83 farms for which soil samples were obtained and processed, no differences in the overall percentage of *R. equi* carrying AMRGs were found between 2017 (median, 0.05%; range, 0–15%) and 2021 (median, 0.03%; range 0–18%; *p*-value = 0.76 [Wilcoxon signed-rank test]; Table 2). The number of farms with *R. equi* carrying AMRGs identified in soil samples was 63% (52/83) in 2017 and 55% (46/83) in 2021, and this difference was not significant (*p*-value = 0.43 [Wilcoxon signed-rank test]; Table 2, Figure 1). Regarding the change in the proportion of *R. equi* carrying AMRGs in the soil from 2017 to 2021, 39% of the farms (32/83) had an increased *R. equi* carrying AMRGs proportion, 36% (30/83) had a decreased proportion, and 25% (21/83) remained free of *R. equi* carrying AMRGs; 11% (9/83) of the farms had an increase in the proportion of *R. equi* carrying AMRGs from zero in 2017, and 18% (15/83) of the farms had a decrease in the proportion of *R. equi* carrying AMRGs to zero in 2021 (Table 2, Figure 1).

### 3.3. Prevalence of Multidrug-Resistant R. equi

The MICs of all the *R. equi* carrying *erm*(51) or *erm*(46) were measured for macrolide, tetracycline, ansamycin, streptogramins, lincosamide, glycopeptide, and aminopyrimidine/sulfonamide antimicrobials (Table 3). The proportion of non-susceptibility to macrolides, tetracycline, clindamycin, and quinupristin–dalfopristin was 100%. Most isolates were resistant to rifampicin (97.5%). For trimethoprim–sulfamethoxazone and doxycycline, most isolates were susceptible (66.25% and 93.7%, respectively). All isolates were susceptible to vancomycin (Table 3). Five (6.25%) isolates were resistant to all antimicrobials tested, except vancomycin.

### 3.4. Prevalence of Antimicrobial Residues

Mass spectrometry was used to measure the concentration of ampicillin, azithromycin, clarithromycin, cycloheximide, doxycycline, erythromycin, sulfonamide, tetracycline, and tylosin in soil samples from 2017 and 2021 from 83 farms. Antimicrobial traces were found for azithromycin, clarithromycin, and erythromycin only. The antimicrobial found with the highest concentrations in soil was erythromycin (mean, 0.11 μg/Kg; median, 0.0 μg/Kg; range, 0.0–1.3 μg/Kg), followed by clarithromycin (mean, 0.06 μg/Kg; median, 0.0 μg/Kg; range, 0.0–1.6 μg/Kg), and azithromycin (mean, 1.8 × 10^−6^ μg/Kg; median, 0.0 μg/Kg; range, 0.0–0.19 μg/Kg).

The combined sum of the macrolide residue amount found for each farm and each time period was used as the primary variable representing antimicrobial residue contamination in the environment and its effect on AMR. The combined amount of macrolide residue had a median of 0.016 μg/Kg with a range of 0.000–1.58 μg/Kg and was not statistically significantly different between 2017 (median, 0.016 μg/Kg; range, 0.002–1.52 μg/Kg) and 2021 (median, 0.013 μg/Kg; range, 0.0–1.58 μg/Kg; *p*-value = 0.08 [Wilcoxon signed-rank test]; Table 4, Figure 2).

### 3.5. Effect of Antimicrobial Use and Year on R. equi Carrying AMRGs

The odds of finding *R. equi* carrying AMRGs at farms increased significantly by 3.6-fold (95% CI, 1.2–10.5, *p*-value = 0.03) when macrolide residue was present in the environment compared to absent. The odds of finding *R. equi* carrying AMRGs in 2021 compared with 2017 was 0.75 (95% CI, 0.3–1.7, *p*-value = 0.45) and not statistically significant (Table 5). Macrolide residue was positively and statistically significantly (estimate, 1.1, SE, 0.5, *p*-value = 0.02) associated with the log-transformed percentage of *R. equi* carrying AMRGs. The year was not statistically significantly associated with the log-transformed percentage of *R. equi* carrying AMRGs (estimate, 0.2, SE, 0.3, *p*-value = 0.48; Table 5, Figure 3).

### 3.6. Effect of Using Thoracic Ultrasound Screening (TUS) and Year on R. equi Carrying AMRGs

The odds of finding *R. equi* carrying AMRGs at a farm increased significantly by 5.4-fold (95% CI, 1.2- to 23.9-fold, *p*-value = 0.03) at farms that used TUS for at least 4 years versus farms that reported not using TUS during the preceding 4 years. The odds of having AMRGs did not change significantly in 2021 compared with 2017 (OR, 0.8; 95% CI, 0.3–2.1; *p*-value = 0.63; Table 6). The use of TUS and the year were not statistically significantly associated with the log-transformed percentage of *R. equi* carrying AMRGs (estimate, 0.7; SE, 0.4, *p*-value = 0.08; and estimate, 0.03; SE, 0.3, *p*-value = 0.99, respectively; Table 6, Figure 4).

### 3.7. Effect of Using Thoracic Ultrasound Screening (TUS) and Year on Antimicrobial Residues

The odds of finding antimicrobial residues were 6.4-fold higher (95% CI, 0.8–52.5-fold; *p*-value = 0.09) at farms that used TUS for at least 4 years versus farms that reported never using TUS or at least not in the last 4 years, but this was not statistically significant. The odds of having antimicrobial residues did not change significantly in 2021 compared with 2017 (OR, 0.4; 95% CI, 0.1–1.5; *p*-value = 0.18; Table 7).

## 4. Discussion

In this study, the presence of macrolide residues was positively associated with the environmental presence of *R. equi* carrying AMRGs at horse-breeding farms. These AMRGs are known to cause resistance to first-line antimicrobials used for the treatment of rhodococcosis in foals. Interest has been growing in investigating the environment as a potential source of selection for AMR and as an important contributor to the global AMR crisis. Recent reports show that antimicrobial pollution drives AMR [30], and the role of the environment as a reservoir for AMR is generally underrepresented in action plans to mitigate AMR [31]. The development of AMR in microbials is considered a natural evolutionary process, but the presence of antimicrobials stimulates the AMR mechanisms of pathogens [32]. Antimicrobials, including macrolides, can persist in the environment for long periods of time, establishing an antimicrobial pressure sufficient to maintain the selection of MDR organisms [33]. Therefore, mitigating AMR without understanding the role of environmental contamination with antimicrobial residues might not be possible.

The environment, including water and soil, provides a vastly diverse AMRGs pool that is higher than the ones found in humans or animal microbiota [34]. Therefore, the increase in pressure due to antimicrobial pollution in the environment promotes opportunities for high levels of novel AMR mechanism acquisition. This is of special importance for pathogenic organisms that live naturally in the soil and cause disease in humans and animals, such as *R. equi*. Thus, studying the effects of antimicrobial pollution in the environment on the AMR evolution of *R. equi* provides a good model for predicting the detrimental long-term effects of overusing antimicrobials in animals and humans. The environment is the main source of infection for foals with *R. equi* [35]. Therefore, the selection of MDR *R. equi* in the environment of horse-breeding farms is likely a strong contributor to the increase in MDR *R. equi* infections seen in foals in recent decades [17]. Because of the potential long-term effect of antimicrobial residues in the environments of microbial communities reported in this study, infections from MDR *R. equi* will likely continue to increase. Apart from the potential long-term selection of MDR *R. equi* and its detrimental effect on equine health, the accumulation of antimicrobial residues establishes constant antimicrobial pressure for the enhancement and spread of MDR pathogens with the potential to threaten humans and other animals through contact or the consumption of contaminated food and water [36]. Previous studies have shown that antimicrobial-resistant *R. equi* have spread internationally [37], demonstrating the potential global impact of even a single event of AMRGs acquisition [38]. Moreover, the accumulation of residues can have ecotoxicological effects, interfering with the normal microbiome of soils and plants and affecting agricultural yield [39]. Therefore, the existence, persistence, and impact of antimicrobial residues in diverse systems and their potential to cause adverse health effects constitute an emerging global crisis, worthy of collaborative, interdisciplinary mitigation efforts.

Studying the association between antimicrobial use and resistance is challenging due to a lack of data on antimicrobial use and potential response bias to questionnaires. Therefore, independent methods to assess antimicrobial pressure in certain settings are needed. In this study, the recovery of data about antimicrobial use in foals from enrolled farms through the collection of questionnaires was limited, but antimicrobial residue measurements in the soil served as an unbiased source of information on existing antimicrobial pressure in the environment of farms. Studies have shown that animals treated with antimicrobials can shed antimicrobial metabolites for weeks after discontinuing therapy, and the antimicrobial residues in the soil can drive the selection of resistant organisms [40]. In our study, residues of several antimicrobials were measured, but traces were only found for macrolides. The accumulation of antimicrobials in the soil depends on many factors, including frequency, dose, the duration of treatment, and shedding rates, as well as degradation processes such as transformation, photodegradation, runoff, and hydrolysis [41,42,43,44,45,46]. Macrolides are frequently used to treat foals in these farms, are excreted in high rates following treatment, and are stable residues known to bind strongly to soil components. Therefore, the persistence of macrolide residues in these farms is likely and expected to be higher than other antimicrobials commonly used. Among macrolides, the erythromycin concentration was highest in the soil samples compared with the clarithromycin and azithromycin concentrations. This result was not surprising because erythromycin is a natural compound found in soil and a fermentation product from *Saccharopolyspora erythraea* [47]. Therefore, its presence in the soil could be attributed to both its natural occurrence and the shedding of this antimicrobial into the soil following the treatment of foals. More studies are needed to investigate the percentage of erythromycin residues that are attributable to antimicrobial use versus natural production. Clarithromycin and azithromycin are, however, semi-synthetic compounds and more stable analogs than erythromycin [48]. Therefore, the presence of these compounds in the soil of horse-breeding farms likely originates from their use to treat and prevent diseases in foals.

Antimicrobial residue was a strong predictor of AMR in the environment at horse-breeding farms in this study, and our results are consistent with the previously reported effects of antimicrobial residues on the amplification of antimicrobial-resistant communities in the environment [30]. Several previous studies have shown irrefutable evidence of the increase in resistant *R. equi*, likely driven by the overuse of antimicrobials in foals [29,49]. Thus, prudent antimicrobial use at horse-breeding farms has been recommended to mitigate AMR and protect foal health. However, reducing antimicrobial use alone might not be sufficient to fight the spread of MDR *R. equi* at horse-breeding farms. Because our data on antimicrobial use in this study were limited, it was not possible to establish if antimicrobial residues persisted over time without antimicrobial use in animals or if the presence of antimicrobials in the environment was due to constant antimicrobial use. However, this snapshot of this strong association between antimicrobial residues and AMR indicates that the management of environments and herds at horse-breeding farms can help control and limit the spread of MDR *R. equi*. Practices, such as delimiting antimicrobial-contaminated areas and avoiding contact with horses, establishing a washout time after antimicrobial therapy is discontinued before animals are reintroduced to the herd, and choosing effective antimicrobials that have shorter half-lives in the environment might contribute to AMR mitigation at horse-breeding farms. However, information on the shedding rates for different antimicrobials and the persistence of their residues in the environment after therapy in foals is unknown. It is also important to note that previous studies have reported environmental factors such as soil texture and mineral components can affect the recovery rate and persistence of antimicrobial residues and AMRGs in the environment [50,51,52]. Therefore, enforcing such practices is not yet indicated because more studies investigating the rate of residue shedding, its persistence, its potential to establish antimicrobial pressure and drive AMR, and if contact with MDR *R. equi* in the soil is a risk factor for its acquisition by foals are needed.

In this study, the presence of macrolide residues, even though below the MIC for *R. equi*, was associated with a higher prevalence of *R. equi* carrying AMRGs. The direct correlation between antimicrobial residues and AMR in the environment is difficult to predict because antimicrobial residues can impose additive and co-selective pressure. A recent study showed that an erythromycin concentration of 2.5 mg/Kg in soil was capable of changing the microbial diversity [53]. This concentration is still much higher than that found in farms in this study. Therefore, we predict that the increased proportion of *R. equi* carrying AMRGs was likely promoted by the facilitation of horizontal gene transfer due to low-level antimicrobial residue exposure. However, more studies are needed to investigate the effect of low-level antimicrobial residue on the microbiome of these farms. Moreover, the bioavailability of antimicrobials in complex matrixes, such as soil, and soil quality regarding the presence of metals can act as co-selective agents [54]. Soil quality was not measured in this study; however, in future studies, these data could provide valuable insights into selective pressure from antimicrobial and non-antimicrobial sources on these farms.

In this study, we found that farms continue to use TUS as a screening method for *R. equi* infections. TUS was previously identified as a risk factor for the presence of MDR *R. equi* in the environment at horse-breeding farms [29] due to the resulting overuse of antimicrobials to prevent clinical signs of the disease. In this study, it was found that TUS continues to be an important risk factor for AMR at horse-breeding farms. Due to the limited access to data about antimicrobial use at the participating farms, it was not possible to establish whether TUS was associated with an increased rate of antimicrobial treatment at farms. However, the use of TUS at farms has been associated with an overuse of antimicrobials in foals in past studies [29]. In this study, we found that the use of TUS may have contributed to the presence of antimicrobial residues in the soil of farms, albeit not significantly (*p*-value = 0.09). Because antimicrobial residues can persist for a long period of time after the initial environmental contamination, farms that have stopped using TUS in recent years but that have historically used it might continue to present levels of residues sufficient to impact microbial selection. This would explain why the current use of TUS is not a strong predictor of antimicrobial residues in our study. Recent studies have shown that the use of TUS is associated with higher rates of antimicrobial use at farms [29] and does not contribute to better survival and recovery of foals subclinically affected by *R. equi* pneumonia [55]. Therefore, the evidence demonstrates that TUS not only has little effect on preventing severe *R. equi* disease in endemic horse-breeding farms, but it may also contribute to the long-term accumulation of antimicrobial residues and the selection of MDR *R. equi* in the environment, increasing the risk of foal infections with MDR *R. equi*.

## 5. Conclusions

We report that antimicrobial residues in the soil are an unbiased measurement of antimicrobial pressure and are strongly associated with the AMR of *R. equi* at horse-breeding farms. Antimicrobial residues might persist in the environment, establishing a continuous selection of MDR *R. equi* in the soil of farms. The historic use of antimicrobials in farms relying on TUS may have set an existing antimicrobial pressure that has persisted over time, contributing to the selection of *R. equi* carrying AMRGs. Given the current prevalence of MDR *R. equi* and antimicrobial residues in the environment of farms, the prevalence of MDR *R. equi* infections in foals will likely continue to increase due to the constant exposure. However, more studies are needed to better predict the effect of the historic use of antimicrobials on the selection of MDR *R. equi* and if the exposure of foals to MDR *R. equi* in the soil is a risk factor for infections with these resistant pathogens.

## Figures and Tables

**Figure 1 vetsci-11-00092-f001:**
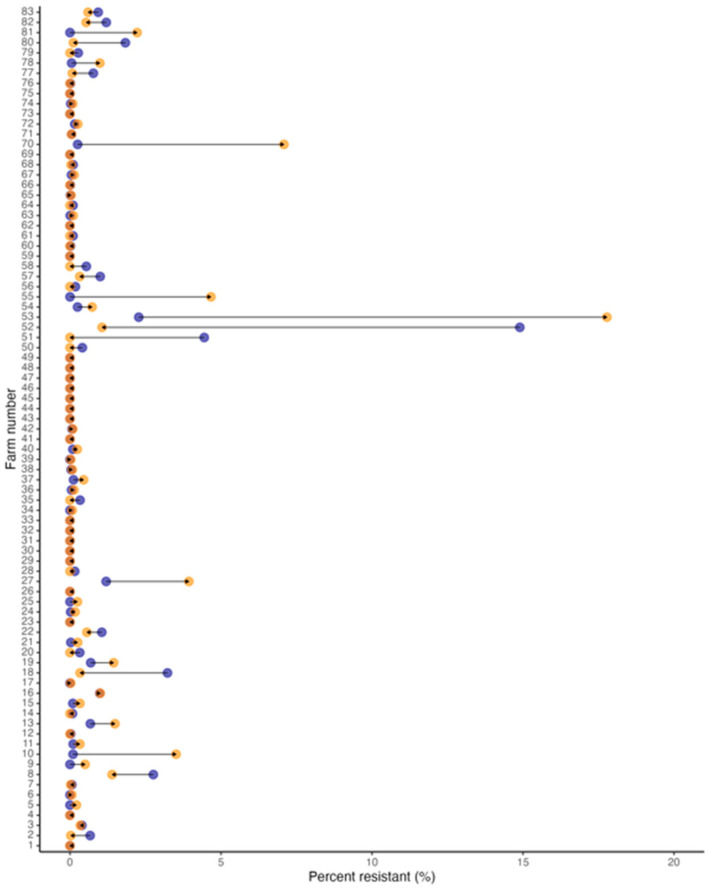
Dumbbell plots with the change in the proportion of resistant MDR *R. equi* in each farm between 2017 (blue) and 2021 (orange); arrows represent directionality from 2017 to 2021.

**Figure 2 vetsci-11-00092-f002:**
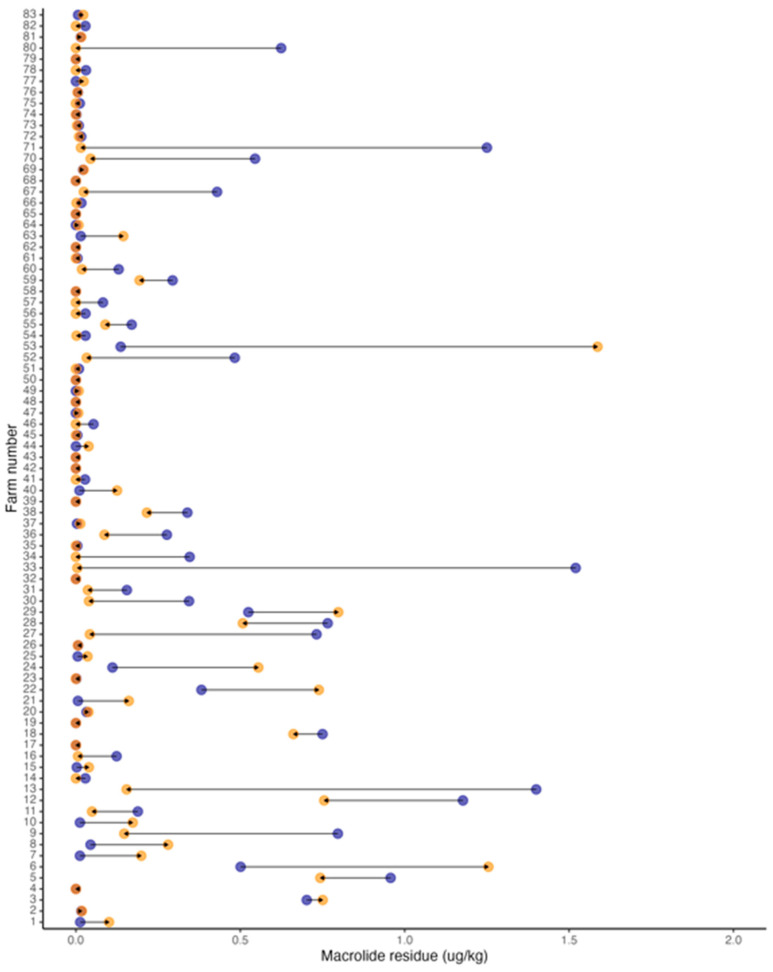
Dumbbell plots with the change in concentration of macrolide residue in the soil from each farm between 2017 (blue) and 2021 (orange); arrows represent directionality from 2017 to 2021.

**Figure 3 vetsci-11-00092-f003:**
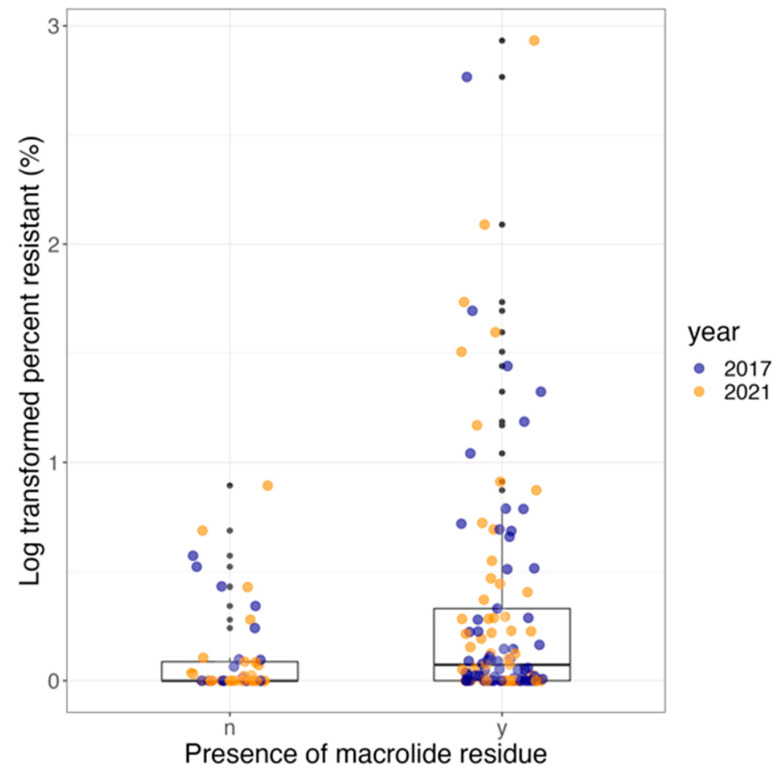
Boxplot showing the proportion of MDR *R. equi* in farms where macrolides were present or absent, color-coded by year: 2017 (blue) and 2021 (orange).

**Figure 4 vetsci-11-00092-f004:**
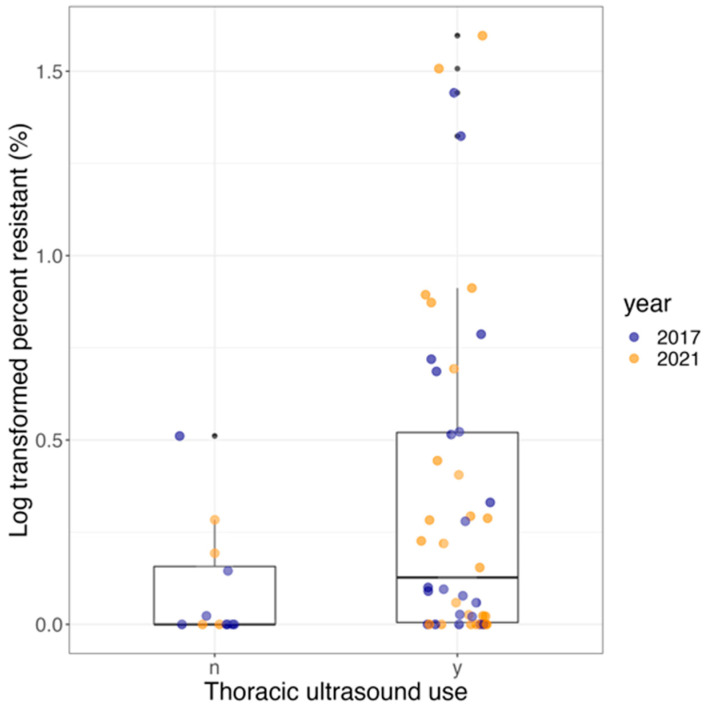
Boxplot showing the proportion of MDR *R. equi* in farms that use or do not use TUS, color-coded by year: 2017 (blue) and 2021 (orange).

**Table 1 vetsci-11-00092-t001:** Descriptive data. Overall antimicrobial treatment, farm density, and mortality in 29 farms from which a full questionnaire was available for both years (2017 and 2021). Percentage of farms using TUS over time from 56 farms from which TUS data were recovered during the second follow-up effort.

Data from Farms with Questionnaire (29 Farms)	2017	2021
Percentage of foals treated with any antimicrobial (median, IQR)	10% (0–20%)	12% (3–27%)
Percentage of foals treated with macrolides (median, IQR)	0% (0–10%)	2% (0–7%)
Proportion of foals treated with rifampicin (median, IQR)	7% (0–19%)	7% (0–18%)
Number of farms that treated foals with any antimicrobials (%)	19/29 (66%)	23/29 (79%)
Number of farms that treated foals with macrolides (%)	11/29 (38%)	20/29 (69%)
Number of farms that treated foals with rifampicin (%)	18/29 (62%)	22/29 (76%)
Density of animals/acre (median, IQR)	0.85 (0.61–1.15)	0.62 (0.44–1.06)
Foal mortality (median, IQR)	0.00 (0–0.00)	0.00 (0–0.002)
**TUS in Farms Over Time (56 farms)**		**N (%)**
Number of farms that used TUS in 2017 Number of farms that used TUS in 2021 Number of farms that used TUS in the last decade Number of farms that did not use TUS in the last decade Number of farms that **did use** TUS from 2014 to 2017 but discontinued use of TUS after 2017 Number of farms that **did not use** TUS from 2014 to 2017 but began use of TUS after 2017		43/56 (77%)
		37/56 (66%)
		32/56 (57%)
		8/56 (14%)
		15/56 (27%)
		1/56 (2%)

**Table 2 vetsci-11-00092-t002:** Descriptive data. The percentage of farms where MDR *R. equi* was found to have decreased or increased over time. For each farm categorization, the median and range percentages of MDR *R. equi* relative to total *R. equi* CFUs are shown.

MDR-*R. equi* in Farms	N (%)	Percentage MDR *R. equi* (Median, Range)
Farms with MDR *R. equi* in 2017	52/83 (63%)	0.05% (0.00–15%)
Farms with MDR *R. equi* in 2021	46/83 (55%)	0.03% (0.00–18%)
Farms with increased MDR *R. equi*	32/83 (39%)	0.30% (0.00–2%)
Farms with decreased MDR *R. equi*	30/83 (36%)	0.40% (0.00–15%)
Farms that remained MDR *R. equi*-free	21/83 (25%)	0.00% (0.00–0.00%)
Farms that increased from zero MDR *R. equi*	9/83 (11%)	0.20% (0.07–0.90%)
Farms that decreased to zero MDR *R. equi*	15/83 (18%)	0.20% (0.05–0.30%)

**Table 3 vetsci-11-00092-t003:** Minimum Inhibitory Concentrations (MIC) for *R. equi* carrying macrolide resistance genes. MIC (median and range) and percent of isolates classified as susceptible or non-susceptible from the total number of isolates tested (N = 80) are shown for each antimicrobial tested with the respective antimicrobial classes they belong to.

Antimicrobial	Class	MIC (Median, Range)	% Susceptible	% Non-Susceptible
Azithromycin	Macrolides	>256 (NA)	0%	100%
Clarithromycin	Macrolides	>256 (NA)	0%	100%
Erythromycin	Macrolides	>256 (24–>256)	0%	100%
Tetracycline	Tetracyclines	8 (4–12)	0%	100%
Doxycycline	Tetracyclines	1 (0.023–>256)	94%	6%
Rifampicin	Ansamycins	>256 (0.032–>256)	2%	98%
Quinupristin-Dalfopristin	Streptogramins	24 (3–>256)	0%	100%
Trimethoprim-sulfamethoxazole	Aminopyrimidines/Sulfonamides	0.75 (0.19–>256)	66%	34%
Vancomycin	Glycopeptides	0.19 (0.025–0.75)	100%	0%
Clindamycin	Lincosamides	>256 (3–>256)	0%	100%

**Table 4 vetsci-11-00092-t004:** Descriptive data. The percentage of farms where macrolide residue was found to have decreased or increased over time. For each farm categorization, the median and range concentrations of macrolide residue are shown.

Antimicrobial Residue in Farms	N (%)	Macrolide Residue (Median, Range; μg/Kg)
Farms with macrolide residue in 2017	64/83 (77%)	0.016 (0.00–1.52)
Farms with macrolide residue in 2021	53/83 (64%)	0.013 (0.00–1.58)
Farms with increased macrolide residue	26/83 (31%)	0.070 (0.020–0.26)
Farms with decreased macrolide residue	43/83 (52%)	0.030 (0.010–0.090)
Farms remained macrolide residue free	14/83 (17%)	0.00 (0.00–0.00)
Farms that increased from zero macrolide residue	5/83 (6%)	0.010 (0.010–0.020)
Farms that decreased to zero macrolide residue	16/83 (19%)	0.030 (0.00–0.040)

**Table 5 vetsci-11-00092-t005:** Effects of year and antimicrobial residue on *R. equi* carrying antimicrobial resistance genes (AMRGs). Estimates, standard errors (SE), and *p*-values generated from a negative binomial mixed-effects model with the log proportion of *R. equi* carrying AMRGs as the outcome and farm identification as a random variable. Odds ratios (OR) and 95% confidence intervals (CI) generated from a logistic mixed-effects model with the presence or absence of *R. equi* carrying AMRGs as the outcome and farm identification as a random variable. * represents *p*-value < 0.05.

Negative Binomial Model	Estimate	SE	*p*-Value
Year 2021 (reference: 2017)	0.22	0.30	0.48
Presence of residue (reference: absence)	1.12	0.51	0.02 *
**Logistic Model**	**OR**	**95% CI**	***p*-value**
Year 2021 (reference: 2017)	0.75	0.33–1.65	0.45
Presence of residue (reference: absence)	3.55	1.19–10.52	0.03 *

**Table 6 vetsci-11-00092-t006:** Effects of year and the use of thoracic ultrasound screening (TUS) on *R. equi* carrying antimicrobial resistance genes (AMRGs). Estimates, standard errors (SE), and *p*-values generated from a negative binomial mixed-effects model with the log proportion of *R. equi* carrying AMRGs as the outcome and farm identification as a random variable. Dataset contains 56 farms that have provided information on TUS use for both periods (2014–2017 and 2018–2021). Odds ratios (ORs) and 95% confidence intervals (CIs) generated from a logistic mixed-effects model with the presence or absence of *R. equi* carrying AMRGs as the outcome and farm identification as a random variable. * represents *p*-value < 0.05.

Negative Binomial Model	Estimate	SE	*p*-Value
Year 2021 (reference: 2017)	0.003	0.34	0.99
Have used TUS at least in the last 4 years (reference: never used TUS or stopped this practice for at least 4 years)	0.73	0.43	0.08
**Logistic Model**	**OR**	**95% CI**	***p*-value**
Year 2021 (reference: 2017)	0.79	0.31–2.05	0.63
Have used TUS at least in the last 4 years (reference: never used TUS or stopped this practice for at least 4 years)	5.43	1.24–23.89	0.03 *

**Table 7 vetsci-11-00092-t007:** Effects of year and the use of thoracic ultrasound screening (TUS) on antimicrobial residues. Odds ratios (ORs) and 95% confidence intervals (CIs) generated from a logistic mixed-effects model with the presence or absence of macrolide residues as the outcome and farm identification as a random variable. Dataset contains 56 farms that have provided information on TUS use for both periods (2014–2017 and 2018–2021).

Logistic Model	OR	95% CI	*p*-Value
Year 2021 (reference: 2017)	0.44	0.14–1.46	0.18
Have used TUS at least in the last 4 years (reference: never used TUS or stopped this practice for at least 4 years)	6.36	0.76–52.52	0.09

## Data Availability

The data presented in this study are available on request from the corresponding author.

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
