# Peer review of "Antimicrobial Residue Accumulation Contributes to Higher Levels of *Rhodococcus equi* Carrying Resistance Genes in the Environment of Horse-Breeding Farms"

_vetsci, 2024, doi:10.3390/vetsci11020092_

Round 1
Reviewer 1 Report
Comments and Suggestions for Authors
Higgins et al. present an insightful study highlighting the relationship between residual antimicrobials and antimicrobial resistance genes and strains in various horse farms in Kentucky, USA. While the authors adopt a localized approach, focusing solely on farms within the state, there is a limitation in the lack of information regarding the use of antimicrobials in the analyzed farms.
However, the paper reveals compelling results, demonstrating a significant association between the use of early diagnosis tools such as TUS and the isolation of MDR-R. equi strains. It addresses crucial aspects of the presence and prevalence of MDR-R. equi strains, a global issue in horse farms. Consequently, the paper holds importance and should be considered for publication after addressing some minor points
It might be beneficial for the authors to include information about the geographical locations of the farms on the map. This additional detail could provide insights into how the environment may influence the development and spread of bacterial strains.
R. equi Quantification: I recommend that the authors perform PCR analysis on some colonies from NANAT (no antibiotics) in addition to the colonies from the antibiotic-treated samples. This additional step will serve to validate their initial morphological analysis
Line 144 Change to capital the “s” of Staphylococcus aureus.
Table 3: use the same decimal format. Rifampicin, don´t use 98 % use 97.50 %.
I recommend that the authors consistently use the same color for the years 2017 and 2021 across all graphs to enhance clarity. In Figure 3, 2017 is represented in blue and 2021 in orange, which is the opposite of Figures 1 and 2.
Author Response
Dear reviewer,
We appreciate the time put into reviewing our paper and making it better.
In respect to mapping the farms: we agree that geographical analyses would be important for this paper, but unfortunately we were not able to perform these analyses due to the agreement we have with the farms to try to maintain their confidentiality.
In respect to the PCR assays: we agree that the validation of the PCR assay is important and the use of susceptible strains to confirm the morphological parameters is essencial. This protocols were developed in our labs in the past years, and protocols were conducted following previously published evidence. Therefore, we didn't repeat these experiments for this paper.
Line 144: the error was corrected, thank you!
Decimals were corrected.
Thank you, the color discrepancies were corrected: The colors are correct in the graphs, but the legends in figure 1 and 2 were incorrect.
Thank you so much for your comments!
Reviewer 2 Report
Comments and Suggestions for Authors
The authors report risk for AMR due to R. equi resistant strains.
Overall, the paper is a little bit not organized, needs an improvement in reporting data both in text and in tables, and be more specular among sections. Not completely undestandable in the general meaning.
Major revison needed.
Summary: in my opinion the summary should be improved adding some more specific concepts, too general in this way.
key words: please add R. equi because the revision is just on R. equi not on other noxae.
line 64-65: please erase the main ... R. equi, it is reduntant.
line 69-71: TUS use for prevention. Please try to better explain for a not expert on R. equi.
line 91: purpose is.
lines 188 and 192: what means 2017 and 2021? the sample collection was performed between 2018 and 2021. Please explain.
line 201-202, 209: the survey has been performed from 2018 to 2021 or just in 2017 and 2021? this must be clarified throughout the sections.
line205-206: did you contact in 2021 or did they not performed the questionnaire in 2021? not clear at all.
table 1: farm were 29 or 56? did ot clear (text and table).
table: treated with please correct all throughout table 1. Sometimens you put % sometimes not. Needs to be improved and corrected in order to uniform tha table.
Farm that did not use tus: you did not say this in the text or not clear, anyway. Please try to be more specular in text and table and add information in text if you put information in tables.
line 225: no differences, you did not say how you assess that no differences exist, the only way are statistical differences and apply a statistical test. You did not report.
The name of the antibiotics needs to be written lowercase.
Author Response
We really appreciate the reviewers comments and the following modifications substantially improve the quality of this paper:
Summary: we edited the summary to be more to the point considering our focus and finding.
R. equi was added as a key-word
The details on the use of TUS were added: "This technique consists in performing TUS in young foals to early identify lung lesions presumptively caused by R. equi and initiate antimicrobial treatment before the development of clinical signs (during the subclinical phase)."
We had 2 objectives in this study; to make it clearer, we added numbering to the objectives: "The purposes of this study are: (i) to compare rates of antimicrobial use, prevalence of R. equi carrying AMRGs, and the concentration of antimicrobial residues at horse-breeding farms in Kentucky over time and (ii) to determine the effect of antimicrobial use and antimicrobial residue accumulation on the prevalence of R. equi carrying AMRGs found in the environment at horse-breeding farms"
lines 188 and 192: what means 2017 and 2021? the sample collection was performed between 2018 and 2021. Please explain. - We apologize for confusing language, the phrase was modified to: "soil samples were thawed.."
Questionnaire collection methodology was clarified by adding: "Questionnaires received as part of the study conducted in 2017 [17] contained information for years 2014 to 2017; and questionnaires collected in 2021 contained information for years 2018 to 2021."
Table 1 legend was rewritten to better explain the number of farms from which questionnaire was recovered. We had 2 attempts of collecting questionnaires in 2021: the first attempt, 29 farms completed the full questionnaire. For the second attempt, we focused only on collecting TUS data and 56 farms provided that information. This explanation is included in the text: "Of the 83 farms included in the study, 86% (71/83) completed a questionnaire in 2017, and 29 out of these 71 farms (41%) completed a questionnaire in 2021. Loss-of-follow-up was dependent on the outcome of antimicrobial use in this study, therefore, questionnaire data are only represented descriptively. Because TUS is a variable of high interest based on previous epidemiological studies [17], [28], we contacted farms that did not originally respond to a questionnaire in 2021 and were able to collect data about TUS practices for further inferential analyses. This second follow-up effort yielded complete paired data for TUS from 56 farms."
For line 225, the methodology to compare groups was added in M&M: "Paired data with non-parametric distributions were compared using Wilcoxon signed-rank test." Statistical significance was reported in: "Of the 83 farms for which soil samples were obtained and processed, no differences in the overall percentage of R. equi carrying AMRGs were found between 2017 (median, 0.05%; IQR, 0-0.3%) and 2021 (median, 0.03%; IQR 0-0.3%; p-value = 0.76; Table 2). "
Names of antibiotics were corrected to start with lowercase.
We appreciate your time to suggest these corrections and improve our manuscript greatly.
Reviewer 3 Report
Comments and Suggestions for Authors
I do not have any comments. In my opinion, the text is ready for publication.
Author Response
We thank the reviewer for taking the time to review this manuscript!
Reviewer 4 Report
Comments and Suggestions for Authors
line 144- Staphylococcus - the first letter of the genus of the bacteria must be capitalized
Author Response
Thank you! We have made the corrections suggested. We appreciate the reviewer taking the time to review our paper.
Reviewer 5 Report
Comments and Suggestions for Authors
The manuscript titled “Antimicrobial residue accumulation contributes to higher levels of Rhodococcus equi carrying resistance genes in the environment of horse-breeding farms” addresses the important issue of the environmental source of antimicrobial resistance and the interactions between the animals and their environment. The subject is important and interesting and fits the scope of the journal. The study was well designed and analyzed. However, the study had some major limitations (as the authors stated themselves) and, therefore, the conclusions are limited. Moreover, the presentation of the data is a little hard to follow and the data is only partially presented, and hard to interpret. In my opinion, it may be published with some modifications.
General comments:
- Throughout the manuscript data is presented as median and IQR. I do suggest adding the range to better represent the data, since it is sometimes difficult to understand. For example, the authors mention that one of the highest residues found was of azithromycin (Line 260), however both the median and IQR are zero, which means that 75% of samples did not have any traces.
- The results are presented separately from the statistical analysis. In some tables these are proportions and rates, but no statistical comparison between groups, while the tables describing the statistics do not compare the rates/proportion/medians etc. between groups. While reading, I kept scrolling back and forward to look at the data and its interpretation. I suggest adding the statistic for each group to the analysis tables for easier read.
- Although residues of several antimicrobials were measured, traces were only found for macrolides. How do the authors explain this, since a variety of antimicrobials have been used in the farms.
- One of the main assumptions of this study was that the use of TUS is associated with the use of antimicrobials (and specifically macrolides, which are used for R. equi infection). However, the data presented in section 3.1 shows an increase in the use of antimicrobials, and specifically macrolides, and a decrease in the use of TUS. How do these to observations concur?
- On the same subject, TUS was found to be significantly associated with the presence of resistant bacteria, but not to the presence of residues. This issue should be further addressed in the discussion (Lines 423-442).
- The authors state that measuring antimicrobial residues in the soil is an unbiased measure (line 369), which may be true, however the association between the type and concentration of these residues and the type and time of antimicrobials used at each farm was not investigated, partly due to limited information. Nonetheless, I do suggest analyzing the associations between residues and what is known regarding the extent, timing and antimicrobial types used at each farm.
Minor corrections:
- Lines 81-83 – Unclear, please rephrase.
- Lines 89-91 – It may not be sufficient to prevent it, but it may help eventually reduce it.
- Line 91 – Please replace “are” with “were”.
- Line 120 – “R. equi” should be in italics.
- Lines 131 and 133 – Please replace “and/or” with “with or without”.
- Line 136 – Please define how R. equi carrying AMRGs were classified as such. Was it sufficient to be positive to either gene, or both?
- Lines 202-203 – “Loss-of-follow-up 202 was dependent on the outcome of antimicrobial use in this study” please clarify.
- Lines 220 and thereafter – please check the correct formal for headlines.
- Line 234 – Table 2 – Please define proportion of what? Of positive samples out of the 27 per farm? From CFUs isolated? It is crucial, since all statistical analysis used this proportion as the outcome.
- I suggest switching the colors representing the years in Figure 3 for consistency with the previous figures.
- Line 241 – In the M&M you mentioned isolating erm(46) and erm(51).
- Line 279 – Was the presence of macrolide residues analyzed as a dichotomous or continues (i.e. concentration) parameter?
- Line 313 – If I understand correctly, the comparison is not between timepoints, but between farms with or without history of using TUS. If so, I suggest replacing “increased” with “…were 6.4-fold higher…”
- Line 316 – Please delete “of”.
- Line 329 – please delete “the” before “antimicrobial pollution”.
- Line 373 – Do the authors have information regarding the use of the different macrolides among equine vets or within the farms? Is erythromycin the most frequently used? Or is the only explanation in the natural excretion? If so, then controlled use of this antimicrobial will not reduce the environmental source of resistance.
- Line 398 – The half life is different within the animal (which influences excretion) and in the environment. Is there a preferable macrolide which should be recommended for the use in horses?
Author Response
We really appreciated the reviewer's comment and we have made all the modifications suggested which has greatly improved our manuscript. Below, please find the specific comments.
General comments:
- Throughout the manuscript data is presented as median and IQR. I do suggest adding the range to better represent the data, since it is sometimes difficult to understand. For example, the authors mention that one of the highest residues found was of azithromycin (Line 260), however both the median and IQR are zero, which means that 75% of samples did not have any traces.
Thank you for this comment, we have modified all the IWR for antimicrobial residues and for MDR-R. equi percentage to range.
- The results are presented separately from the statistical analysis. In some tables these are proportions and rates, but no statistical comparison between groups, while the tables describing the statistics do not compare the rates/proportion/medians etc. between groups. While reading, I kept scrolling back and forward to look at the data and its interpretation. I suggest adding the statistic for each group to the analysis tables for easier read.
This is a really good point. We have added the statistics test to the text when applicable.
- Although residues of several antimicrobials were measured, traces were only found for macrolides. How do the authors explain this, since a variety of antimicrobials have been used in the farms.
Thank you for this comment. More discussion was added around this finding: “In our study, residues of several antimicrobials were measured but traces were only found for macrolides. Accumulation of antimicrobials in the soil depend on many factors, including frequency, dose, duration of treatment, and shedding rates, as well as degradation processes such as transformation, photodegradation, runoff, and hydrolysis. Macrolides are frequently used to treat foals in these farms, to be excreted in high rates following treatment and are stable residues, known to bind strongly to soil components. Therefore, the persistence of macrolides residues in these farms is likely and expected to be higher than other antimicrobials commonly used.”
- One of the main assumptions of this study was that the use of TUS is associated with the use of antimicrobials (and specifically macrolides, which are used for R. equi infection). However, the data presented in section 3.1 shows an increase in the use of antimicrobials, and specifically macrolides, and a decrease in the use of TUS. How do these to observations concur?
That was a surprising fact for us as well. We tried to not focus too much on the antimicrobial use data because these were only for 29 farms. Therefore, we tried to use only the TUS data for further inferential analysis.
- On the same subject, TUS was found to be significantly associated with the presence of resistant bacteria, but not to the presence of residues. This issue should be further addressed in the discussion (Lines 423-442).
This is a great question. More discussion was added: “Because antimicrobial residues can persist for long period of time after the initial environmental contamination, farms that have stopped using TUS in the last years, but that have historically used it, might continue to present levels of residues sufficient to impact microbial selection. This would explain why the current use of TUS is not a strong predictor of antimicrobial residues in our study.”
- The authors state that measuring antimicrobial residues in the soil is an unbiased measure (line 369), which may be true, however the association between the type and concentration of these residues and the type and time of antimicrobials used at each farm was not investigated, partly due to limited information. Nonetheless, I do suggest analyzing the associations between residues and what is known regarding the extent, timing and antimicrobial types used at each farm.
We agree with the stated. Ideally, we would have all variables (antimicrobial use, use of TUS, residues, and AMR) for all farms. However, we had a high rate of loss of follow up in this study and the loss of follow up was associated with the outcome (i.e., AMR was associated with questionnaire response). Therefore, it was not possible to conduct analyses using the questionnaire data, due to the selection bias introduced in our study. In order to mitigate that lack of data on the farm practices, we interview farms one more time to try to collect data on the use of TUS and were able to recover 56 farms. For inferential purposes, we only used the TUS use data from these 56 farms. This is the main limitation of our study, but it is not something we could fix – the remaining farms were not willing to respond to questionnaires.
Minor corrections:
- Lines 81-83 – Unclear, please rephrase.
Thank you, we have rephrased to: “Since then, investigators have shown that foals that are subclinically affected by R. equi recover spontaneously without the need for antimicrobial treatment.”
- Lines 89-91 – It may not be sufficient to prevent it, but it may help eventually reduce it.
Thankyou, that is a good point. We have rephrased it to: “If the persistence of R. equi carrying AMRGs and antimicrobials in the environment occurs at endemic farms, MDR-R. equi infections in foals will likely continue to occur.”
- Line 91 – Please replace “are” with “were”.
Thank you, corrections were made.
- Line 120 – “R. equi” should be in italics.
Thank you, corrections were made.
- Lines 131 and 133 – Please replace “and/or” with “with or without”.
Thank you, corrections were made.
- Line 136 – Please define how R. equi carrying AMRGs were classified as such. Was it sufficient to be positive to either gene, or both?
Thank you, this detail was added as: “R. equi were classified as carrying AMRGs if positive to either erm(46) or erm(51).”
- Lines 202-203 – “Loss-of-follow-up 202 was dependent on the outcome of antimicrobial use in this study” please clarify.
Rephrased to: “Loss-of-follow-up was associated with the presence of R. equi carrying AMRGs, therefore, questionnaire data are only represented descriptively.”
- Lines 220 and thereafter – please check the correct formal for headlines.
Thank you!
- Line 234 – Table 2 – Please define proportion of what? Of positive samples out of the 27 per farm? From CFUs isolated? It is crucial, since all statistical analysis used this proportion as the outcome.
Thank you. It has been specified further with: “For each farm categorization, the median and range percentage MDR-R. equi relative to total R. equi CFU is shown”
- I suggest switching the colors representing the years in Figure 3 for consistency with the previous figures.
Thank you, the legends for figures 1 and 3 were actually wrong, so we fixed that without having to replace the figures.
- Line 241 – In the M&M you mentioned isolating erm(46) and erm(51).
Thank you! It was a typo.
- Line 279 – Was the presence of macrolide residues analyzed as a dichotomous or continues (i.e. concentration) parameter?
For these analysis, we did both. But I see the confusion, I cited the wrong table for the dichotomous results. I have now corrected to indicate table 5.
- Line 313 – If I understand correctly, the comparison is not between timepoints, but between farms with or without history of using TUS. If so, I suggest replacing “increased” with “…were 6.4-fold higher…”
- Line 316 – Please delete “of”.
- Line 329 – please delete “the” before “antimicrobial pollution”.
- Line 373 – Do the authors have information regarding the use of the different macrolides among equine vets or within the farms? Is erythromycin the most frequently used? Or is the only explanation in the natural excretion? If so, then controlled use of this antimicrobial will not reduce the environmental source of resistance.
- Line 398 – The half life is different within the animal (which influences excretion) and in the environment. Is there a preferable macrolide which should be recommended for the use in horses?